# Comparative Analysis of Iodine Levels, Biochemical Responses, and Thyroid Gene Expression in Rats Fed Diets with Kale Biofortified with 5,7-Diiodo-8-Quinolinol

**DOI:** 10.3390/ijms26020822

**Published:** 2025-01-19

**Authors:** Justyna Waśniowska, Ewa Piątkowska, Piotr Pawlicki, Sylwester Smoleń, Aneta Kopeć, Agnieszka Dyląg, Joanna Krzemińska, Aneta Koronowicz

**Affiliations:** 1Department of Human Nutrition and Dietetics, Faculty of Food Technology, University of Agriculture in Krakow, al. Mickiewicza 21, 31–120 Krakow, Poland; justyna.wasniowska@student.urk.edu.pl (J.W.); ewa.piatkowska@urk.edu.pl (E.P.); aneta.kopec@urk.edu.pl (A.K.); agnieszka.dylag@student.urk.edu.pl (A.D.); joanna.krzeminska@urk.edu.pl (J.K.); 2Department of Basic Sciences, Faculty of Veterinary Medicine, University of Agriculture in Krakow, Redzina 1c, 30–248, Krakow, Poland; piotr.pawlicki@urk.edu.pl; 3Department of Plant Biology and Biotechnology, Faculty of Biotechnology and Horticulture, University of Agriculture in Krakow, al. Mickiewicza 21, 31–120 Krakow, Poland; sylwester.smolen@urk.edu.pl

**Keywords:** curly kale, biofortification, iodine, 5,7-diiodo-8-quinolinol, iodine metabolism, rats

## Abstract

Iodine is a key micronutrient essential for the synthesis of thyroid hormone, which regulates metabolic processes and maintains overall health. Despite its importance, iodine deficiency is a global health issue, leading to disorders such as goiter, hypothyroidism, and developmental abnormalities. Biofortification of crops with iodine is a promising strategy to enhance the dietary iodine intake, providing an alternative to iodized salt. Curly kale (*Brassica oleracea* var. *sabellica*) is a nutrient-rich vegetable high in vitamins A, C, K; minerals; fiber; and bioactive compounds with antioxidant, anti-inflammatory, and detoxifying properties. This study evaluates the effects of diets containing iodine-biofortified curly kale (‘Oldenbor F_1_’ and ‘Redbor F_1_’) on iodine content, tissue iodine levels, and various biochemical parameters in laboratory rats. The biofortified curly kale was enriched with 5,7-diiodo-8-quinolinol. The iodine content in the AIN-93G (control) diet and the non-biofortified curly kale diets did not differ significantly. However, diets with 5,7-diiodo-8-quinolinol biofortified kale showed significantly higher iodine levels compared with the control diets. Tissue analysis revealed the highest iodine concentrations in the liver and kidneys of rats fed diets with biofortified curly kale, indicating better iodine bioavailability. Biochemical analysis showed that rats fed the biofortified kale diet had lower total cholesterol (TC) and triglyceride (TG) levels compared with rats fed the control diet. Additionally, the biofortified diet improved the liver function markers (ALAT, ASAT) and reduced oxidative stress markers (TBARS). The study also investigated the expression of thyroid-related genes (*Slc5A5*, *Tpo*, *Dio1*, *Dio2*) in response to diets containing biofortified kale. The results demonstrated significant changes in gene expression, indicating adaptive mechanisms to dietary iodine levels and the presence of bioactive compounds in the biofortified kale. The study also observed variations in uric acid levels, with lower concentrations in rats fed a diet with biofortified curly kale. Biofortified curly kale supports thyroid function and improves liver and kidney health by reducing oxidative stress and modulating key biochemical and genetic markers. These findings suggest that biofortified curly kale can effectively increase dietary iodine intake as a nutritional intervention to address iodine deficiency and promote overall health.

## 1. Introduction

Iodine is a key micronutrient required for the synthesis of thyroid hormones, which play an important role in regulating metabolic processes and maintaining overall health. Despite its importance, iodine deficiency remains a significant global health problem, leading to disorders such as goiter, hypothyroidism, and various developmental abnormalities. One strategy to address this is to seek alternatives to iodizing table salt. One such alternative is the biofortification of plants with iodoquinolines. It may become an effective strategy to increase the iodine content of foods, and thus, improve the dietary iodine intake [1,2].

Curly kale (*Brassica oleracea* var. *sabellica*) is a nutrient-rich cruciferous vegetable known for its high content of vitamins (A, C, K), minerals (calcium, iron), fiber, and bioactive compounds such as glucosinolates, flavonoids, and carotenoids [3,4,5,6]. These compounds possess antioxidant, anti-inflammatory, and detoxifying properties, rendering kale a valuable addition to a healthy diet. By biofortifying curly kale with iodine in the form of 5,7-diI-8-Q, it is possible to exploit its health benefits while supplementing dietary iodine deficiency [3,7,8].

The introduction of biofortification of vegetables with iodoquinoline compounds such as 5,7-diI-8-Q represents a novel approach to addressing iodine deficiency on a global scale. Biofortification is a process of nutritional enrichment of crops that can significantly improve dietary quality, especially in regions where micronutrient deficiencies are common. Studies have shown that biofortified kale not only provides essential nutrients, but also effectively increases iodine levels in in the body.

Furthermore, the use of 5,7-diI-8-Q in the biofortification process is a pioneering step, as this chemical compound has not been extensively studied in the context of plant iodine fortification. It is hypothesized that 5,7-diI-8-Q may occur in plants [9]. However, the safety of consuming plants fortified with this compound should be determined by in vivo studies. Its efficacy and safety as a biofortification agent represent an interesting area of research that may open up new opportunities in the field of nutrition and public health. Future research could focus on optimizing the biofortification processes and assessing the long-term effects of consuming such vegetables and their impact on human health, thus contributing to more sustainable and nutritious food systems.

Iodine metabolism and maintenance, other than thyroid functions, depend on the subtleties of molecular mechanisms interplaying at the expression level of key genes involved with iodide transport, synthesis, and activation of hormones. Of these, the gene *Slc5a5* encodes sodium/iodide symporter (*NIS*), which is responsible for iodide transport into thyroid follicular cells and the initial event in thyroid hormone biosynthesis. The *Tpo* gene also codes for thyroid peroxidase, an enzyme that catalyzes the organification of iodide and the coupling of iodotyrosines, two steps that are critical to thyroid hormone synthesis. The *Dio1* and *Dio2* genes encode deiodinase enzymes, which either inactivate or activate thyroid hormones through their control of availability and activity at the tissue level. Although the function of these genes in the thyroid is well described, their expression under specific dietary conditions, such as on diets supplemented with biofortified compounds, is still poorly explored [10,11,12].

The present study was designed to evaluate the effects of diets containing biofortified curly kale (‘Oldenbor F_1_’ and ‘Redbor F_1_’) on the dietary iodine content, tissue iodine levels, and various biochemical parameters in laboratory rats. The biofortified kale used in this study was enriched with 5,7-diiodo-8-quinolinol (5,7-diI-8-Q), a compound that is a source of iodine. Comparisons were made with control diets, including the standard AIN-93G diet and diets containing non-biofortified curly kale (‘Oldenbor F_1_’ and ‘Redbor F_1_’). Moreover, the study analyzed the expression of genes related to iodine metabolism and thyroid function to assess how biofortified kale affects iodine transport, metabolism, and utilization. This approach was necessary to determine whether biofortified kale not only increases iodine levels but also supports thyroid hormone synthesis and normal physiological processes, allowing for a comprehensive evaluation of its effectiveness in treating iodine deficiency.

## 2. Results

### 2.1. Iodine Content in Rats’ Diets

The iodine contents of the AIN-93 G (C, control) diet and diets with curly kale (‘Oldenbor F_1_’, ‘Redbor F_1_’) without biofortification were not significantly different. The same was true of diets with curly kale (‘Oldenbor F_1_’, ‘Redbor F_1_’) with biofortification 5,7-diI-8-Q. Significant differences in iodine content were found between the diets with kale (‘Oldenbor F_1_’, ‘Redbor F_1_’) with 5,7-diI-8-Q biofortification and the control diet AIN-93 G and diets with curly kale (‘Oldenbor F_1_’, ‘Redbor F_1_’) without biofortification. The iodine content of the C, CO, and CR diets was lower than that of the BO and BR diets (Appendix A).

### 2.2. Iodine in Selected Tissues

The kidney and liver tissue iodine contents were affected by various dietary treatments (Table 1). The highest tissue iodine levels, e.g., liver and kidney, were found in rats fed diets containing biofortified kale ‘Oldenbor F_1_’ and ‘Redbor F_1_’ (BO diet and BR diet) compared with other experimental groups. Furthermore, the lowest iodine contents (in the liver) were determined in animals fed the C, CO, and CR diets compared with rodents fed diets with biofortified kale (BO and BR diets). A significant difference was observed in the iodine content of the kidneys of rats fed the C diet compared with those fed the CR diet.

### 2.3. Selected Biochemical Parameters

Rats fed the BO vs. CO and BR vs. CR diets did not show significant differences within a kale variety with respect to selected biochemical parameters, i.e., total cholesterol (TC), low-density lipoprotein (LDL), very low-density lipoprotein (VLDL), triglyceride (TG), high-density lipoprotein (HDL), alanine aminotransferase (ALT), aspartate aminotransferase (AST), thiobarbituric acid reactive substances (TBARS), total bilirubin (BIL TOTAL), direct bilirubin (BIL DIREC), uric acid (UA), glutathione reductase (GR), total antioxidant status (TAS), triiodothyronine (T3), thyroxine (T4), and thyroid-stimulating hormone (TSH). On the other hand, significant differences were found for the parameters ALT (BO vs. CO) and BIL DIREC (BR vs. CR).

The rats that were fed the diet containing biofortified kale ‘Redbor F_1_’ (BR diet) had a significantly lower concentration of total cholesterol (TC) in their serum than the rats that were fed the control diet (C diet). In the other experimental groups, no significant differences were observed (Table 2). It was also found that animals fed the control diet (C diet) had significantly higher serum HDL concentrations than rats fed the BO, CR, or BR diets. (Table 2). The level of LDL was not affected by the various dietary treatments (Table 2). The lowest concentration of TG was measured in the serum of rats fed the BR diet compared with the serum of rodents fed the C diet.

The highest concentration of ALT was measured in the serum of rats fed the C diet compared with rodents fed the BO, CR, and BR diets. The highest concentration of AST was measured in the serum of rats fed the C diet compared with rodents fed the BR diet (Table 2). The highest concentration of TBARS was measured in the serum of rats fed the C diet compared with rodents fed the CR diet (Table 2). The highest concentration of BIL TOTAL was measured in the serum of rats fed the C diet compared with the other experimental groups. The lowest concentration of BIL TOTAL was measured in the serum of rats fed the BO diet compared with rats fed the C diet (Table 2). No significant differences in BIL DIRECT concentrations were observed between the control diet (C) and the CO, BO, CR, and BR diets (Table 2).

The lowest concentration of UA was measured in the serum of rats fed the BR diet compared with rodents fed the C diet. No significant differences were observed in the concentration of the parameter between the other groups.

The levels of GR, TAS, T3, and T4 were not affected by the various dietary treatments compared with rodents fed the C diet. (Table 2). The highest concentration of TSH was measured in the serum of rats fed the C diet compared with rodents fed the CO and BO diets (Table 2).

### 2.4. Evaluation of Tpo, Slc5a5, Dio1, and Dio2 Gene Expression Among Experimental Groups

(Figure 1a–d) Expression levels of thyroid-related genes (*Tpo*, *Slc5a5*, *Dio1*, *Dio2*) were analyzed using digital PCR (dPCR) and expressed as the number of copies per microliter. Type of diet the rats were fed: C, control diet (AIN-93G); CO, diet containing control curly kale ‘Oldenbor F_1_’ without biofortification; BO, diet containing biofortified curly kale ‘Oldenbor F_1_’; CR, diet containing control curly kale ‘Redbor F_1_’; BR, diet containing biofortified curly kale ‘Redbor F_1_’. Values in rows with different letters (a, b, c, d) are significantly different, *p* ≤ 0.05 (one-way analysis (ANOVA), standard error (*n* = 8)).

(Figure 1a) The highest expression of *Tpo* mRNA was observed in the CO group (*p* < 0.05) in comparison with all other study groups. This group exhibited significant differences from the C, BO, CR, and BR groups. In the control group (C), *Tpo* expression was observed to be lower than in the CO group. However, it was significantly higher than in the BO (*p* < 0.05), CR (*p* < 0.05), and BR (*p* < 0.05) groups. The BO group exhibited *Tpo* mRNA levels comparable to those observed in the CR group, with no statistically significant difference between the two (*p* > 0.05). However, both groups demonstrated higher *Tpo* gene expression than the BR group. The BR group, with the lowest mean *Tpo* expression (~300 copies/μL), differed significantly from the C, CO, and CR groups (*p* < 0.05).

(Figure 1b) The highest expression of *Slc5a5* mRNA was observed in the CO group (*p* < 0.05) in comparison with all other groups. The CO group exhibited a statistically significant divergence from the C, BO, CR, and BR groups. The control group (C) exhibited a significantly lower *Slc5a5* expression than the CO group, yet a higher expression than the other groups (*p* < 0.05). The *Slc5a5* mRNA expression in the CR group was approximately 35 copies/μL, which was significantly higher than in the BO and BR groups (*p* < 0.05). The BO group exhibited a significantly lower expression level than the C, CO, and CR groups (*p* < 0.05) while displaying a higher level than the BR group (*p* < 0.05). The lowest mRNA expression of the *Slc5a5* gene was observed in the BR group, with a value of approximately 20 copies/μL. This was significantly lower compared with all other groups (*p* < 0.05).

(Figure 1c) The highest levels of mRNA *Dio1* gene were observed in the CO group (*p* < 0.05), which exhibited a significant divergence from all other study groups. In the CR group, *Dio1* expression was observed to be lower than in the CO group yet significantly higher in comparison with the C, BO, and BR groups (*p* < 0.05). The *Dio1* gene expression observed in the C, BO, and BR groups was found to be similar and not significantly different between the groups (*p* > 0.05). Nevertheless, all these groups exhibited significantly reduced expression levels in comparison with the CR and CO groups (*p* < 0.05).

(Figure 1d) The highest mRNA level of the *Dio2* gene was observed in the CO group (*p* < 0.05) in comparison with all other study groups, with a mean value of approximately 3.5 copies/μL. The aforementioned group exhibited significant divergence from the C, BO, CR, and BR groups. The second highest expression level was observed in the BR group, which was significantly higher than that observed in the C, BO, and CR groups (*p* < 0.05). The control group (C) exhibited an average *Dio2* gene expression of approximately 1.8 copies/μL, which was significantly lower than that observed in the CO, BO, and BR groups (*p* < 0.05). The mean *Dio2* gene expression in the BO group was approximately 2 copies/μL, which was higher than that in the C and CR groups (*p* < 0.05). The lowest *Dio2* gene expression was observed in the C and CR groups. These were significantly lower compared with all other study groups (*p* < 0.05).

## 3. Discussion

Kale (*Brassica oleracea* var. *sabellica*) is one of the most prized cruciferous vegetables. It owes its exceptional nutritional and health properties to its high content of fiber, minerals (e.g., calcium, potassium), vitamins (e.g., C, E, K), and bioactive compounds (e.g., glucosinolates, flavonoids, polyphenols), which have detoxifying, anti-inflammatory, and antioxidant properties [13,14]. Biofortification, the process of increasing micronutrient levels in crops, is becoming an increasingly popular approach to addressing micronutrient deficiencies worldwide. Kale biofortified with iodine can provide an innovative and effective solution to iodine deficiency, combining the health benefits of consuming this vegetable with an additional source of iodine [7,15].

The majority of iodine in the body is present in the thyroid tissues, comprising approximately 70–80% of the total iodine content. The remaining iodine is found in the kidneys, liver, and muscles [16]. Our results showed that the highest iodine concentration in the selected organs (liver, kidney) was observed in rats fed a diet with the addition of biofortified curly kale. It is likely that this was due to the increased iodine content of these diets. The results demonstrated that the highest concentration of iodine was present in the kidneys (Table 1). This is due to the metabolic function of this organ. The kidneys are vital organs that perform a number of essential functions within the body. These include the filtration of blood, the removal of metabolic by-products, the regulation of electrolyte and acid–base balance, the control of blood pressure, and the support of red blood cell production. The precise and complex regulation of these processes is essential for maintaining homeostasis and the health of the entire body [17,18]. There are only a few studies that show changes in the iodine concentration in the tissues. Piątkowska et al. [19] also showed that the kidneys contained the highest concentration of iodine compared with the other tissues analyzed (liver, heart, thyroid) in rats fed a diet with carrots (raw and cooked) KI-enriched compared with rats fed a control diet. In contrast, Kopeć et al. [20] conducted a study on rats fed diets with biofortified KI lettuce, a control diet, and a diet containing non-biofortified KI control lettuce. They showed that the liver and thigh muscles had the highest tissue levels of this trace element in rats fed the biofortified lettuce. In Hou et al. [21], the study revealed that iodine concentrations can be high in healthy adult individuals’ hair and skin. According to some authors, iodine contained in various foods, including fortified products, has a high bioavailability of about 99% [22,23]. In our study, we also observed that iodine levels in different organs as well as in urine and feces (citing our previous studies [24]) depended on dietary iodine levels.

The results demonstrated that rats fed a diet containing biofortified ‘Redbor F_1_’ kale (BR diet) showed a significant reduction in serum total cholesterol (TC) concentrations in comparison with rats on the control diet (Table 2). However, the LDL + VLDL level was not affected by the different diet treatments. We observed reduced triglycerides levels in rats fed the BR diet compared with those fed the C diet. This may be due to the presence of fiber and other biologically active ingredients such as flavonoids, a group of health-promoting compounds. When consumed, they prevent the oxidation of low-density cholesterol (LDL) fraction, thereby counteracting the formation of atherosclerotic deposits [25]. Another ingredient is tocopherol, a group of vitamin E, which among other things affects the elasticity of blood vessels, contributing to the lowering of serum lipids [26,27]. The higher levels of anthocyanins in ‘Redbor F_1_’ may have contributed to the reduction in TC and TG levels. Research suggests that the consumption of anthocyanin-rich foods may contribute to a reduction in total cholesterol levels. This may be due to a reduction in cholesterol synthesis in the liver and an increase in the excretion of bile acids, which are derivatives of cholesterol. Anthocyanins can affect gene expression and the activity of the enzymes involved in lipid synthesis [28,29,30,31,32]. In particular, they can reduce the activity of enzymes such as diacylglycerol acyltransferase (DGAT), which plays a key role in triglyceride synthesis. In the study, other researchers showed that the total cholesterol level was not affected by different diets, which is supported by the cited studies discussed in the next sentence [19,20]. In addition, Kopeć et al. [20] showed that in the serum of rats fed biofortified KI or control lettuce, the concentration of TC and LDL + VLDL cholesterol increased significantly. However, Piątkowska et al. [19] showed that the level of LDL + VLDL tended to decrease in the serum of rats fed a diet containing iodine biofortified (KI) raw carrots. Additionally, we discovered that the level of HDL has a tendency to decrease in the serum of rats fed BO, CR, and BR diets, but this parameter was still within physiological norms in the serum [20]. Interestingly, von Lintig demonstrated that a diet rich in β-carotene may reduce non-HDL cholesterol levels, thereby lowering the risk of atherosclerosis and cardiovascular diseases. This suggests that bioactive compounds present in vegetables including kale could exert a beneficial effect on lipid metabolism and cardiovascular health [33].

ALT activity decrease was affected by the BO, CR, and BR diets. No significant differences were observed in serum AST levels between the four experimental groups (CO, BO, CR, BR). The decrease in TBARS activity was influenced by the CR and BR diets (Table 2). The reduction in the above parameters may be due to the bioactive compounds found in kale [34]. Sulforaphane, a compound in cruciferous vegetables, may protect the liver from oxidative and inflammatory damage, lowering ALT and AST levels [35,36]. Kale also contains flavonoids like quercetin and kaempferol, which contribute to its health benefits. They have antioxidant, anti-inflammatory, and anti-fibrotic effects and help maintain liver health by reducing levels of oxidative stress markers such as TBARS [37,38]. The reduction in the abovementioned parameters may be due to the form of iodine enrichment. 5,7-diiodo-8-quinolinol is a quinoline derivative in which one hydroxyl group (-OH) and two iodine groups (-I) are attached to the quinoline ring. Compounds in this group have various biological and pharmacological activities that are beneficial to health, such as anti-inflammatory, immunomodulatory, antibacterial, and antifungal activities. It is possible that the properties of this group, as well as iodine, play a role in lowering ALT, AST, and TBARS parameters. On the other hand, researchers using vegetables enriched with a different form of iodine (KI) have obtained different results from ours. In an animal study, liver parameters (ALT, AST) were shown to increase [19] or not change after a diet of iodine-fortified (KI) vegetables [19,20]. On the other hand, the TBARS parameter also decreased after diets with biofortified (KI) vegetables (raw carrots and lettuce) [19,20]. The use of iodine-rich herbs such as seaweed in the Gao et al. study showed less damage to thyroid follicular cells compared with excess iodine alone. The study showed that the activity of antioxidant enzymes such as the oxidative stress marker malondialdehyde (MDA) decreased in the iodine-rich herb group. This suggests that iodine-rich plants may reduce oxidative stress in the thyroid [39].

In our study, we also found that total bilirubin levels decreased in the curly kale-fortified diets. The greatest reduction was seen in the BO diet with curly kale biofortified with 5,7-diI-8-Q compared with the C control group. The glucosinolates found in kale, which are metabolized to active isothiocyanates such as sulforaphane after consumption, may have contributed to these results. These compounds have potent detoxifying properties that can support liver function by inducing phase II detoxification enzymes. Sulforaphane activates enzymes such as glutathione S-transferase (GST) and NQO1 (NAD(P)H-oxidoreductase). These enzymes help to neutralize toxins and metabolites, which may support liver health and indirectly affect bilirubin metabolism [40,41,42]. In contrast to the above results, no significant differences were observed between the control group and the groups fed the curly kale diet without biofortification and with biofortified 5,7-di-8-Q (BR). One potential explanation for the observed variations could be attributed to the reduced iodine content of the kale that was not biofortified. Conversely, in the BR group, where the kale was biofortified with the compound 5,7-di-8-Q, the variety of kale may also have exerted an influence on the outcomes. However, it should be emphasized that this is merely a hypothesis that will be substantiated upon the completion of comprehensive compositional analyses of both kale varieties. These analyses are currently underway, and the results will be disseminated in the ensuing scientific publication.

The results indicate a variation in uric acid levels between different groups, with statistical significance indicating notable differences between the group fed a diet based on AIN-93G and the group fed a diet based on BR, which included curly kale biofortified with 5,7-di-8-Q. The higher uric acid levels in rats fed the standard AIN-93G diet compared with those fed the 5,7-di-8-Q-enriched curly kale may be due to the high protein and refined carbohydrate content of the AIN-93G diet, which promotes uric acid production. A population-based study conducted in China demonstrated that as urinary iodine concentration increased, blood uric acid level diseases decreased. The underlying mechanism may involve the ability of iodine, in the form of iodide ions, to inhibit renal reabsorption of uric acid by the URAT1 transporter. This inhibition leads to a reduction in blood uric acid levels. However, further research is needed to fully understand this relationship [43]. In contrast, the beneficial properties of curly kale, including antioxidants, fiber, and iodine from 5,7-di-8-Q, help to reduce and regulate uric acid levels more effectively [44,45].

The levels of glutathione reductase and total antioxidant status were not affected by the different dietary treatments (Table 2). This result demonstrates that diets containing biofortified curly kale did not impair the body’s antioxidant defense mechanisms, suggesting that the biofortified kale enriched with 5,7-diiodo-8-quinolinol (5,7-diI-8-Q) does not increase oxidative stress. Maintaining good antioxidant capacity is crucial when introducing biofortified foods to prevent potential oxidative damage caused by the increased iodine intake. These findings indicate that biofortified kale can safely increase iodine levels and support thyroid function without negatively impacting redox balance [46,47,48,49].

Adding biofortified curly kale (‘Oldenbor F_1_’, ‘Redbor F_1_’) to experimental diets did not affect the T3 and T4 levels in the serum of experimental rats, which is an important finding of our study. It can be ascertained that the 5,7-di-8-Q-enriched curly kale, and thus the high iodine content, did not exert a deleterious effect on the health parameters of the rats. In groups of rats fed with kale of the variety ‘Oldenbor F_1_’ (non-biofortified and biofortified), a decrease in TSH was observed (Table 2). In contrast, no significant changes were observed in the ’Redbor F_1_’ variety. Although curly kale is rich in bioactive compounds that may support overall thyroid health, there is no direct evidence that specific compounds in kale lower TSH levels. Similar results were obtained by Piątkowska et al. [19], who showed that biofortified carrots (KI) had no effect on T4 levels, while, in contrast to our study, they caused an increase in TSH and T3. In the study conducted by Hussein et al. [50], it was observed that in rats with thiocyanate-induced hypothyroidism, long-term consumption of excess iodine resulted in a significant increase in T3 and T4 levels.

The findings of the present study indicate that biofortified kale with 5,7-di-8-Q (BO and BR groups) has a notable impact on the expression of crucial genes associated with thyroid function. The genes in question are *Slc5a5*, *Tpo*, *Dio1,* and *Dio2*. A comparison with the control group (C) and the addition of kale without biofortification, CO (from Oldenbor F_1_) and CR (from Redbor F_1_), indicates the presence of compensatory mechanisms in response to excess iodine.

The expression of genes in the thyroid, including *Slc5a5*, *Tpo,* and *Dio1*, was observed to be lowest in the BO and BR groups. Statistical analysis confirmed that these differences were significant (*p* < 0.05), suggesting a clear impact of excess dietary iodine on the activity of these genes. This finding suggests a clear effect of excess iodine in the diet on the activity of these genes. In a study conducted by Hussein et al. [50], the administration of excess iodine (3000 mg/L and 6000 mg/L) was found to result in a significant decrease in the expression of the *Dio1*, *Tpo,* and *Slc5a5* genes in the thyroid gland. A similar outcome was documented in a study conducted by Liang et al. [51], which demonstrated that an excess of iodine (750 µg/d) influenced the diminished expression of *Slc5a5* and *Tpo* in the thyroid gland of rats. An excess of iodine induces rat plasma iodine concentrations above 20–35%, a Wolff–Chaikoff effect, which results in decreased expression of these genes [52]. This is a protective mechanism that prevents excessive iodine accumulation in the thyroid gland. These findings do not suggest adverse effects on the thyroid gland, as evidenced by the findings of Ling et al. [51], who demonstrated that a dose of iodine 100 times the norm is toxic to the body only. The high content of glucosinolates and their metabolites in kale may also have influenced the inhibition of gene activity. Conversely, kale’s bioactive compounds, such as isothiocyanates and indole-3-carbinol, may modulate inflammatory and antioxidant pathways by affecting the regulation of these genes [53]. The elevated expression of *Dio2* in the BO and BR groups in comparison with the control group (C) may represent a compensatory mechanism in response to the diminished activity of the other thyroid genes (*Slc5a5*, *Tpo,* and *Dio1*), which are involved in iodine uptake and thyroid hormone synthesis. It is possible that the thyroid gland increases the activity of *Dio2* to provide sufficient active T3 under conditions of reduced thyroid hormone production [54].

The CO group (green control kale) exhibited the highest expression values for all the genes tested, which may indicate a dietary iodine deficiency or the adverse effect of goitrogens found in kale, thereby hindering its bioavailability. Our findings align with those previously reported by Lavado-Autric et al., who demonstrated that iodine deficiency leads to an increased expression of *Dio1* and *Dio2* mRNA in the thyroid gland as a compensatory mechanism [55]. This compensatory response underscores the critical role of iodine in maintaining thyroid hormone homeostasis. Without adequate iodine, thyroid hormone synthesis is impaired, leading to alterations in metabolic regulation, including lipid and protein metabolism, thermogenesis, and growth processes [3]. The data presented indicate that a diet deficient in iodine induces an increase in *Dio1* mRNA in the thyroid gland. However, the CR group exhibited a markedly lower level of expression than the CO group, despite the absence of 5,7-di-8-Q in kale. These observations necessitate further investigation through a comprehensive analysis of kale composition.

## 4. Materials and Methods

### 4.1. Plant Material

Curly kale cultivation and biofortification with 5,7 diiodo-8-quinolinol was described in our previous publication [24].

### 4.2. Animal Study

The feeding experiment with laboratory rats was described in our previous publication [24]. It was conducted with the permission of the First Local Ethical Committee in Krakow (Poland, resolution no. 568/2021) on five groups of animals (each consisting of *n* = 8). The duration of the experiment was eight weeks.

The experimental diets were prepared based on the AIN-93G diets [24] and were prepared by the company Zoolab (Sędziszów, Poland). The detailed compositions of the diets can be found in Appendix A.

Group 1 was fed the AIN-93G (C) diet, which contained the recommended iodine levels for curly kale as outlined by Reeves [56]. Group 2 (CO) was fed the control diet of curly kale ‘Oldenbor F_1_’, while Group 4 (CR) was fed the control diet of curly kale ‘Redbor F_1_’. Both diets contained a mineral mixture with iodine. In the diet containing biofortified curly kale (group 3, BO diet with biofortified 5,7-diI-8-Q raw curly kale ‘Oldenbor F_1_’ and group 5, BR diet), the only source of iodine was kale (the mineral mixture did not contain iodine) (Appendix A). The only source of iodine in the diet containing biofortified curly kale (group 3, BO diet with biofortified 5,7-diI-8-Q raw curly kale ‘Oldenbor F_1_’) was kale (the mineral mixture did not contain iodine) (Appendix A).

After an 8-week experimental period, fasted (12 h) rats were anaesthetized (substance used—isoflurane 4%; inhaled). Blood was obtained by cardiac puncture and collected in plain test tubes without anticoagulant. Blood samples were left at room temperature for about 2 h to clot. Blood samples were centrifuged (1500× *g*, 15 min) to obtain serum. Hearts, thyroids, livers, and kidneys were dissected, washed in 0.9% sodium chloride, dried with laboratory tissue paper, and weighed. Serum and tissue samples were kept frozen at −80 °C until further analysis.

### 4.3. Analysis of Iodine in Animal Tissues

The analysis was based on research published by Krzeminska et al. [57].

### 4.4. Analysis in Serum and Blood

The serum was analyzed in order to determine the concentration of total cholesterol (TC) (cat. no. Liquick Cor-CHOL 60 2–204, PZ Cormay S.A., Łomianki, Poland), high-density lipoprotein cholesterol (HDL cholesterol) (cat. no. Cormay HDL-2–053, PZ Cormay S.A., Łomianki, Poland), and triacylglycerols (TGs) (cat. no. Liquick Cor-TG 60 2–253, PZ Cormay S.A., Łomianki, Poland). The differences between TC and HDL were employed in the calculation of the LDL + VLDL (low-density lipoprotein + very low-density lipoprotein) level [58]. The activity of aspartate aminotransferase (ALT) and alanine aminotransferase (AST) in the serum was determined using the Liquick Cor-ALAT 60 1–216, PZ Cormay S.A., Łomianki, Poland and Cor-ASAT 60 1-214, PZ Cormay S.A., Łomianki, Poland, respectively. In the serum, the level of thiobarbituric acid reactive substances (TBARS) was measured as previously reported by Ohkawa et al. [59]. The results were shown as nmol of malondialdehyde (MDA) per mL. Serum was analyzed to determine total bilirubin (BIL TOTAL) (cat. no. Liquick Cor-BIL TOTAL 60 2-245, PZ Cormay S.A. Łomianki), Poland), and direct bilirubin (BIL DIRECT) (cat. no. Liquick Cor-BIL DIRECT MALLOY-EVELYN 60 2-348, PZ Cormay S.A. Łomianki). Uric acid (UA) was also determined (Liquick Cor-UA 60 2-208, PZ Cormay S.A. Łomianki). The parameters such as glutathione reductase (GR) and total antioxidant status (TAS) were determined using the following kits: Manual/RX Monza GR 2368, Randox Laboratories Ltd., Crumlin, UK and Manual NX 2332, Randox Laboratories Ltd. UK, respectively. The concentrations of triiodothyronine (T3) and thyroxine (T4) were quantified using the ELISA Kit (cat no. CEA453Ge 96 Test; CEA452Ge 96 Test; respectively, Cloud-Clone Corp, Houston, TX, USA). The level of thyroid-stimulating hormone (TSH) was quantified with the Rat Thyroid Stimulating Hormone ELISA (cat no. RTC007R, BioVendor–Laboratorní medicína a.s, Brno, Czech Republic).

### 4.5. Gene Expression Profiling

#### 4.5.1. RNA Extraction and cDNA Synthesis

Total RNA was extracted from 15 mg of frozen rat thyroid tissue using the AllPrep DNA/RNA Mini Kit (Qiagen, Hilden, Germany, Cat. No. 80204) following the manufacturer’s instructions. The RNA concentration and purity were assessed using a NanoDrop™ 2000/2000c Spectrophotometer (cat. no. ND-2000, Thermo Fisher Scientific, Waltham, MA, USA). All samples exhibited A260/A280 ratios ≥ 1.9 and A260/A230 ratios ≥ 2, indicating high purity. Subsequently, 200 ng of RNA from each sample was reverse-transcribed into cDNA in a 20 µL reaction using the High-Capacity cDNA Reverse Transcription Kit (cat. no. 4368814, Thermo Fisher Scientific, Waltham, MA, USA), following the manufacturer’s protocol: 25 °C for 10 min, 37 °C for 120 min, and 85 °C for 5 min. After the reverse transcription reaction, the cDNA concentration was reassessed using the NanoDrop™ 2000/2000c.

#### 4.5.2. Digital PCR Analysis

Gene expression analysis was conducted using digital PCR (dPCR) with TaqMan^®^ probes specific for the target genes: *Slc5a5* (Rn00583900_m1, Thermo Fisher Scientific, Waltham, MA, USA), *Dio1* (Rn00572183_m1, Thermo Fisher Scientific, Waltham, MA, USA), *Dio2* (Rn00581867_m1, Thermo Fisher Scientific, Waltham, MA, USA), and *Tpo* (Thermo Fisher Scientific, Rn00571159_m1, Waltham, MA, USA). Each dPCR reaction had a total volume of 12 µL, containing QIAcuity Probe Mastermix (cat. no. 250102, QIAGEN, Hilden, Germany), 200 nM of the respective TaqMan^®^ probe, and 2 µL of cDNA template (100 ng). The concentration and purity of cDNA were assessed using a NanoDrop™ 2000/2000c spectrophotometer. Reactions were performed in a QIAcuity Nanoplate 8.5k 24-well (cat. no. 250011, QIAGEN, Hilden, Germany) using the QIAcuity One system (cat. no. 911001, QIAGEN, Hilden, Germany), which partitions the sample into approximately 8500 individual reactions per well, facilitating absolute quantification through Poisson statistics. The thermal cycling conditions were as follows: initial denaturation at 95 °C for 2 min, followed by 40 cycles of denaturation at 95 °C for 15 s and annealing/extension at 60 °C for 15 s. Negative controls, consisting of reactions without cDNA template, were included to monitor for contamination. Each experimental group comprised four biological replicates, with each replicate consisting of pooled material from two individuals. Absolute transcript levels were initially calculated in copies per microliter of PCR. Data analysis was conducted with the QIAcuity Suite software. This methodology ensures high sensitivity and accuracy, eliminating the need for standard curves and minimizing the impact of potential PCR inhibitors.

### 4.6. Statistical Analysis

Data were presented as mean ± SD. One-way analysis of variance (ANOVA) was used to test for differences at *p* ≤ 0.05. Differences between the experimental groups were tested using the Duncan test. Comparative analysis of *Slc5a5*, *Tpo*, *Dio1*, and *Dio2* gene copy numbers between the control group and 4 experimental groups (*n* = 8 per group) was performed using Mann–Whitney test. Results are presented as means ± standard deviation (SD). In the graphs, statistical significance is indicated by letters, where *p* < 0.05. Statistical analysis was performed in triplicate using Statistica 13.1 PL (Dell Inc., Tulsa, OK, USA) and JASP software (Jeffreys’s Amazing Statistics Program, version 0.19.1) to ensure rigorous evaluation of the data.

## 5. Conclusions

Curly kale biofortified with 5,7-di-8-Q (‘Oldenbor F_1_’, ‘Redbor F_1_’) may help regulate uric acid levels in the body owing to its antioxidant content.

In light of the aforementioned findings, future research should concentrate on the long-term effects of biofortified kale consumption on thyroid function and metabolic health. It is necessary to conduct human clinical trials to assess the effectiveness of biofortified kale on body iodine levels and health parameters such as cholesterol, triglycerides, and liver enzymes. It is also essential to analyze the bioavailability of iodine from biofortified kale and to compare it with other iodine sources in order to assess the full potential of this method in improving public health.

The biofortified kale, which has an excess of iodine, has been observed to significantly reduce the expression of genes that are related to thyroid function (*Slc5a5*, *Tpo*, *Dio1*, *Dio2*). This indicates that there are compensatory mechanisms that act in response to the excess iodine and that there is a potential impact of the bioactive compounds such as glucosinolates.

## 6. Patents

The method of biofortification of vegetables in iodine cultivated using a traditional, soilless, and hydroponic method, and the use of 5,7-diiodo-8-quinolinol for biofortification of vegetables with iodine, are covered under patent application number P.443221 for the compounds (Polish Patent Office; 21 December 2022).

## Figures and Tables

**Figure 1 ijms-26-00822-f001:**
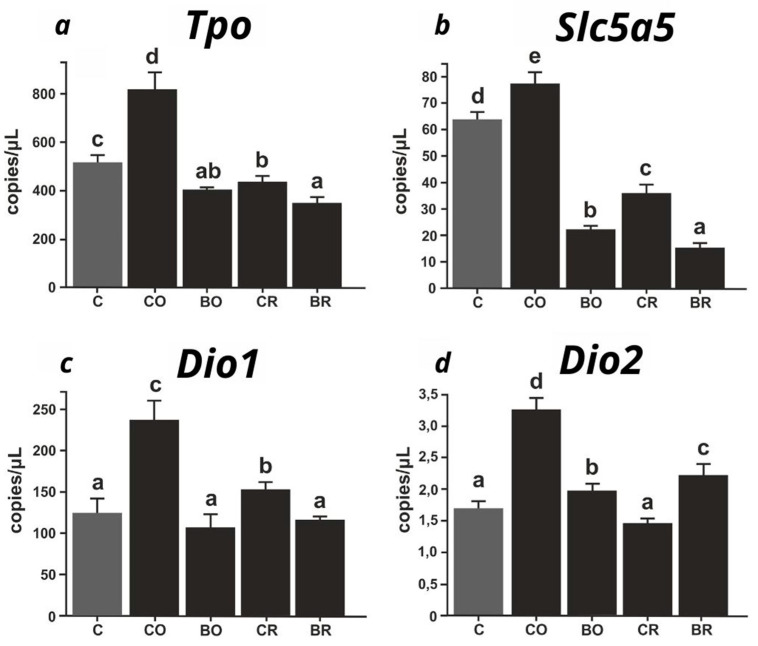
Digital PCR (dPCR) analysis of the expression levels of thyroid-related genes (*Tpo*, *Slc5a5*, *Dio1*, *Dio2*) in experimental rat groups: (**a**) *Tpo* (thyroid peroxidase), (**b**) *Slc5a5* (sodium/iodide symporter), (**c**) *Dio1* (type 1 deiodinase), and (**d**) *Dio2* (type 2 deiodinase). Gene expression levels are presented as copies/µL. Groups are represented as follows: C, control diet (AIN-93G); CO, diet containing control curly kale ‘Oldenbor F_1_’ without biofortification; BO, diet containing biofortified curly kale ‘Oldenbor F_1_’; CR, diet containing control curly kale ‘Redbor F_1_’; BR, diet containing biofortified curly kale ‘Redbor F_1_’. Different letters above bars (e.g., a, b, c, d, e) indicate statistically significant differences (*p* < 0.05) between groups.

**Table 1 ijms-26-00822-t001:** Iodine content in kidneys and livers of experimental rats.

Iodine Content	C	CO	BO	CR	BR
liver mg/kg d.m.	0.12 ^a^ ± 0.01	0.11 ^a^ ± 0.01	0.15 ^b^ ± 0.00	0.11 ^a^ ± 0.00	0.14 ^b^ ± 0.01
kidney mg/kg d.m.	0.14 ^b^ ± 0.00	0.13 ^ab^ ± 0.01	0.19 ^c^ ± 0.01	0.11 ^a^ ± 0.01	0.17 ^c^ ± 0.01

Type of diet fed to rats from which tissues were collected: C, control diet (AIN-93G); CO, diet containing control curly kale ‘Oldenbor F_1_’ without biofortification; BO, diet containing biofortified curly kale ‘Oldenbor F_1_’; CR, diet containing control curly kale ‘Redbor F_1_’; BR, diet containing biofortified curly kale ‘Redbor F_1_’. ‘Oldenbor F_1_’, ‘Redbor F_1_’. Values in rows with different letters (a, b, c) are significantly different, *p* ≤ 0.05 (one-way analysis (ANOVA), standard error (*n* = 0.01)). ^a^ Values with the letter “a” do not differ significantly from each other. ^b^ Values with the letter “b” are significantly different from those with the letter “a” but do not differ from other values labeled “b”. ^c^ Values with the letter “c” are significantly different from those with both “a” and “b”.

**Table 2 ijms-26-00822-t002:** Selected biochemical parameters in blood serum of experimental rats.

	C	CO	BO	CR	BR
TC mmol/L	3.01 ^b^ ± 0.19	2.81 ^ab^ ± 0.16	2.61 ^ab^ ± 0.19	2.61 ^ab^ ± 0.14	2.48 ^a^ ± 0.11
LDL + VLDL mmol/L	1.12 ^a^ ± 0.13	1.07 ^a^ ± 0.12	1.06 ^a^ ± 0.13	1.07 ^a^ ± 0.08	0.93 ^a^ ± 0.11
TG mg/Dl	113.75 ^b^ ± 16.53	122.92 ^b^ ± 16.99	103.90 ^ab^ ± 12.36	83.44 ^ab^ ± 12.56	64.72 ^a^ ± 6.44
HDL mmol/L	1.89 ^b^ ± 0.06	1.78 ^ab^ ± 0.11	1.56 ^a^ ± 0.09	1.54 ^a^ ± 0.10	1.57 ^a^ ± 0.11
ALT U/L	10.13 ^b^ ± 3.02	8.20 ^b^ ± 1.00	2.64 ^a^ ± 0.27	3.96 ^a^ ± 0.43	2.62 ^a^ ± 0.26
AST U/L	12.18 ^b^ ± 2.18	8.00 ^ab^ ± 3.00	8.95 ^ab^ ± 0.60	8.97 ^ab^ ± 0.87	4.74 ^a^ ± 0.55
TBARS nmol/mL	554.52 ^b^ ± 9.67	535.31 ^ab^ ± 31.40	518.56 ^ab^ ± 27.46	474.93 ^a^ ± 22.34	469.96 ^a^ ± 15.85
BIL TOTAL mg/dL	0.60 ^c^ ± 0.13a	0.19 ^ab^ ± 0.04	0.14 ^a^ ± 0.04	0.38 ^b^ ± 0.09	0.22 ^ab^ ± 0.05
BIL DIRECT mg/dL	1.34 ^ab^ ± 0.24	1.50 ^ab^ ± 0.43	0.62 ^a^ ± 0.19	0.80 ^a^ ± 0.28	1.89 ^b^ ± 0.28
UA mg/dL	2.80 ^b^ ± 0.37	2.23 ^ab^ ± 0.41	1.95 ^ab^ ± 0.17	1.89 ^ab^ ± 0.29	1.72 ^a^ ± 0.26
GR U/L	375.72 ^a^ ± 72.74	348.31 ^a^ ± 61.69	391.91 ^a^ ± 43.02	378.96 ^a^ ± 52.54	331.12 ^a^ ± 50.32
TAS mmol/L	0.88 ^a^ ± 0.07	0.86 ^a^ ± 0.03	0.92 ^a^ ± 0.03	0.98 ^a^ ± 0.03	0.95 ^a^ ± 0.04
T3 ng/mL	3.81 ^ab^ ± 0.08	3.92 ^b^ ± 0.06	3.75 ^ab^ ± 0.06	3.68 ^a^ ± 0.04	3.85 ^ab^ ± 0.02
T4 ng/mL	2.75 ^a^ ± 0.22	2.87 ^a^ ± 0.17	2.26 ^a^ ± 0.25	2.59 ^a^ ± 0.42	2.48 ^a^ ± 0.20
TSH ng/mL	2.18 ^b^ ± 0.06	2.02 ^a^ ± 0.04	2.00 ^a^ ± 0.04	2.10 ^ab^ ± 0.05	2.09 ^ab^ ± 0.05

Type of diet the rats were fed: C, control diet (AIN-93G); CO, diet containing control curly kale ‘Oldenbor F_1_’ without biofortification; BO, diet containing biofortified curly kale ‘Oldenbor F_1_’; CR, diet containing control curly kale ‘Redbor F_1_’; BR, diet containing biofortified curly kale ‘Redbor F_1_’. Values in rows with different letters (a, b, c) are significantly different, *p* ≤ 0.05 (one-way analysis (ANOVA), standard error (*n* = 8)). ^a^ Values with the letter “a” do not differ significantly from each other. ^b^ Values with the letter “b” are significantly different from those with the letter “a” but do not differ from other values labeled “b”. ^c^ Values with the letter “c” are significantly different from those with both “a” and “b”.

## Data Availability

The original contributions presented in the study are included in the article, further inquiries can be directed to the corresponding authors.

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
