# Peer review of "Comparative Analysis of Iodine Levels, Biochemical Responses, and Thyroid Gene Expression in Rats Fed Diets with Kale Biofortified with 5,7-Diiodo-8-Quinolinol"

_ijms, 2025, doi:10.3390/ijms26020822_

Round 1
Reviewer 1 Report
Comments and Suggestions for Authors
All comments, suggestions, and questions are available throughought the manuscript.

Author Response
Response to Reviewers' Comments
Title manuscript: Comparative analysis of iodine levels and biochemical responses, and thyroid gene expression in rats fed diets with kale biofortified with 5,7-diiodo-8-quinolinol
General Response
We sincerely thank the reviewers for their thorough evaluation of our manuscript and for providing insightful comments and constructive suggestions. We have carefully addressed each of the comments, as detailed below. Below is a point-by-point response to each comment.
Reviewer 1
Comment 1:
Please authors, give the main example of minerals.
Response:
Thank you for your valuable suggestion. We agree that including more examples would enhance the clarity of the abstract. However, due to the character limit imposed by the journal for abstracts, we are unable to expand this section further.
Comment 2:
Please authors, give some example of them.
Response:
Thank you for your valuable suggestion. We agree that including more examples would enhance the clarity of the abstract. However, due to the character limit imposed by the journal for abstracts, we are unable to expand this section further.
Comment 3:
Please authors, summarize your conclusion on curly kale fortified by 5,7-diiodo-8-quinolinol for tyroide, liver and kidney health improving.
Response:
Thank you for your suggestion. We have summarized our conclusion on the effects of curly kale biofortified with 5,7-diiodo-8-quinolinol on thyroid, liver, and kidney health in lines 37-41 of the updated manuscript.
Comment 4:
Please authors, you can add, the body of the consumers.
Response:
Thank you for your suggestion. After careful consideration, we believe that the phrase "in the body" is the most appropriate choice. It ensures clarity and applies to both human and animal models without causing ambiguity. Using "consumer" could introduce confusion, particularly in the context of experimental animals, whereas "in the body" maintains a neutral and scientifically accepted wording.
Comment 6:
Please authors, you can comment the availabelity of the 5,7-diiodo-8-qinolinol to familiar agriculture.
Response:
Thank you for your insightful comment. We acknowledge that the availability and practical application of 5,7-diiodo-8-quinolinol (5,7-diI-8-Q) in agricultural practices are important considerations for future use of biofortified crops. Currently, this compound is predominantly used in research settings – line 6-70
Comment 7:
0.12a±0.01
Response:
Thank you for your valuable suggestion. 0.12a±0.01 changed correctly to
Comment 8:
Please authors, to all abbreviation write in extensive form to the first time report in the text.
Response:
Thank you for your valuable suggestion. We have ensured that all abbreviations are written in their extensive form when first mentioned in the text. These changes can be found in lines 130-136 of the updated manuscript.
Comment 9:
Please authors, give some example to these substances.
Response:
Thank you for your valuable suggestion. We have added specific examples of minerals, vitamins, and bioactive compounds present in curly kale in lines 225-226 of the updated manuscript
Comment 10:
Please authors, cite these informations.
Response:
Thank you for your valuable suggestion. Text quote: “These include filtration of the blood, removal of metabolic by-products, regulation of electrolyte and acid-base balance, control of blood pressure and support of red blood cell production.” is “[17,18]”-line 243
Comment 11:
Pleas authors, what is the quantity of tocopherol in kale?
Response:
Thank you for your valuable question. The quantification of tocopherol in kale will be provided in our next planned publication. The tocopherol content will be determined as part of a detailed metabolomic analysis using ultra-high-performance liquid chromatography coupled with mass spectrometry (UHPLC-MS). We appreciate your interest and will ensure that these results are shared in due course.
Comment 12:
Please authors, what is the quantity of anthocyanins?
Response:
Thank you for your valuable question. The quantification of anthocyanins in kale will be provided in our next planned publication. The anthocyanins content will be determined as part of a detailed metabolomic analysis using ultra-high-performance liquid chromatography coupled with mass spectrometry (UHPLC-MS). We appreciate your interest and will ensure that these results are shared in due course.
Comment 13:
Please authors, you can report type of diets and cite more studies.
Response:
Thank you for your valuable suggestion. We have added a clarification in line 275 indicating that the description of the referenced studies is provided in the following sentence. However, it is important to note that, to the best of our knowledge, there is currently limited research available on biofortified foods, which explains the relatively small number of studies cited in this area. Additionally, we have added a clarification in line 274 indicating that the description of the referenced studies is provided in the following sentence.
Comment 14:
Please authors, what can you comment about beta-carotene influence and not only about raw carrot consumption?
Response:
Thank you for your valuable suggestion. The sentence has been corrected- line 281-285.
Comment 15:
Please authors, cite these bioactive compounds.
Response:
Thank you for your comment. The citation you referred to is indeed present in line 265 the original version of the manuscript. However, the referenced sentence concludes in line 264, and the citation : ”The reduction in the above parameters may be due to the bioactive compounds found in kale.” has been automatically moved to the following line by the formatting system. Now quoting is in line 290.
Comment 16:
Please authors, you can summarize these sentences.
Response:
Thank you for your valuable suggestion. The sentence has been corrected- line 290-292
Comment 17:
On the other hand, other
Response:
Thank you for your valuable suggestion. The sentence has been corrected- line 301
Comment 18:
Please authors, can you explaine well your novel fidings compared with other studies.
Response:
Thank you for your valuable comment. The paragraph has been completely revised to provide more clarity and detail. The updated version can be found in lines 335-328 of the manuscript.
Comment 19:
Please authors, can you comment if iodine is correlated with reducing uric acid production?
Response:
Thank you for your valuable suggestion. We have added a comment regarding the correlation between iodine and uric acid production in lines 335-340 of the updated manuscript.
Comment 20:
Please authors, report the iodine quantity.
Response:
Thank you for your valuable suggestion. We have included the reported iodine quantity in line 375 of the updated manuscript.
Comment 21:
Please authors, do the same.
Response:
Thank you for your valuable suggestion. We have included the reported iodine quantity in line 378 of the updated manuscript.
Comment 22:
Please authors, do the same.
Response:
Thank you for your valuable suggestion. We have included the reported iodine quantity in line 379-380 of the updated manuscript.
Comment 23:
Please authors, clarify your findigs.
Response:
Thank you for your valuable suggestion. The sentence has been corrected- line 396-399
Comment 24:
Please authors, it is important to improve discussion on iodine importance.
Response:
Thank you for your valuable suggestion. The sentence has been corrected- line 499-402

Reviewer 2 Report
Comments and Suggestions for Authors
The paper "
Comparative analysis of iodine levels and biochemical responses, and thyroid gene expression in rats fed diets with kale biofortified with 5,7-diiodo-8-quinolinol"
conducted by Justyna Waśniowska et. al. presents important informations regarding posibilitatea cresterii aportului de iod alimentar prin biofortificarea culturilor de varza cu iod, oferind o alternativă la sarea iodată.
This study evaluates the effects of diets containing biofortified curly kale with iodine, on tissue iodine levels and various biochemical parameters in laboratory rats.
Tissue analysis revealed better iodine bioavailability, demonstrated by the presence of iodine in higher concentrations in the organs (liver, kidney) of rats fed with biofortified curly kale.
An important aspect highlighted by this research is related to the presence of lower levels of total cholesterol (TC) and triglycerides (TG) in these rats, compared to the control group.
In addition, the biofortified diet improved liver function markers (ALAT, ASAT), reduced oxidative stress markers (TBARS), and led to significant changes in the expression of thyroid-related genes.
Also, the use of 5,7-diI-8-Q in the biofortification process is a pioneering, as this chemical compound has not been extensively studied in the context of iodine fortification of plants. This compound is safe and effective, opening new opportunities in the field of nutrition and public health
The research results could have a significant impact on future dietary strategies for patients with metabolic disorders and could recommend biofortified curly kale as a functional food. This could improve dietary iodine intake, with a major impact on public health by preventing iodine deficiency.
The formulated conclusions are in accordance with the obtained results, based on the proposed study objectives.
The references are recent and relevant to the information presented.
To improve the quality of the publication, I would suggest the following:
1.Check whether the references in the text that refer to figures (such as Figure 1a.)) are correct.
2.Specify the meaning of the letters a, b, c in tables. Specify the meaning of each, even if it indicates statistically significant differences.
Author Response
Response to Reviewers' Comments
Title mamuscript: Comparative analysis of iodine levels and biochemical responses, and thyroid gene expression in rats fed diets with kale biofortified with 5,7-diiodo-8-quinolinol
General Response
We sincerely thank the reviewers for their thorough evaluation of our manuscript and for providing insightful comments and constructive suggestions. We have carefully addressed each of the comments, as detailed below. Below is a point-by-point response to each comment.
Reviewer 2
Comment 1:
Check whether the references in the text that refer to figures (such as Figure 1a.)) are correct.
Response:
Thank you for your comment. We have carefully reviewed all references to figures throughout the manuscript to ensure they are correctly labeled and correspond to the appropriate figures (e.g., Figure 1a).
Comment 2:
Specify the meaning of the letters a, b, c in tables. Specify the meaning of each, even if it indicates statistically significant differences.
Response:
Thank you for your valuable suggestion. The letters a, b, and c in the tables indicate statistically significant differences between groups, as determined by one-way ANOVA followed by a post hoc test (P ≤ 0.05). Specifically:
a: Values with the letter "a" do not differ significantly from each other.
b: Values with the letter "b" are significantly different from those with the letter "a" but do not differ from other values labeled "b."
c: Values with the letter "c" are significantly different from those with both "a" and "b."

Reviewer 3 Report
Comments and Suggestions for Authors
Overall, the manuscript describes a well-designed study and is well written. A few edits/revisions are outlined below with the majority of the revisions required in the discussion.
Abstract
Line 17 – “the iodine deficiency” should be changed to “iodine deficiency” by removing “the”
Discussion
Discussion is generally well written but it is lengthy and contains a significant amount of information regarding biological activity of unfortified kale and other vegetables.
Line 306 – This paragraph, while interesting, does not contribute much to the goal of the study as it is written. Can this be expanded to tie it to the goals of the study?
Line 322 – “witch” needs to be changed to “with”
Line 328 – first sentence needs to be expanded; for example, were these differences in the expression of genes significant?
Materials and Methods
Line 378 – how long were the rats fasted?
Line 379 – I assume “plain” means tubes without anticoagulant?
Line 380 – additional information regarding how the blood was handled should be added, for example, how long was the sample allowed to clot? Was it left at room temperature to clot or on ice? Was it centrifuged at room temperature?
Author Response
Response to Reviewers' Comments
Title: Comparative analysis of iodine levels and biochemical responses, and thyroid gene expression in rats fed diets with kale biofortified with 5,7-diiodo-8-quinolinol
General Response
We sincerely thank the reviewers for their thorough evaluation of our manuscript and for providing insightful comments and constructive suggestions. We have carefully addressed each of the comments, as detailed below. Below is a point-by-point response to each comment.
Reviewer 3
Comment 1:
Line 17 – “the iodine deficiency” should be changed to “iodine deficiency” by removing “the”
Response:
Thank you for your valuable suggestion. We have removed "the" in line 17, changing the phrase to "iodine deficiency" as recommended.
Comment 2:
Line 306 – This paragraph, while interesting, does not contribute much to the goal of the study as it is written. Can this be expanded to tie it to the goals of the study?
Response:
Thank you for your valuable suggestion. We have expanded the paragraph to better align it with the goals of the study, as recommended. The revised paragraph can be found in lines 343-350 of the updated manuscript.
Comment 3:
Line 322 – “witch” needs to be changed to “with”
Response:
Thank you for pointing out this error. We have corrected "witch" to "with" in line 364 as suggested.
Comment 4:
Line 328 – first sentence needs to be expanded; for example, were these differences in the expression of genes significant?
Response:
We are grateful for your constructive feedback, which has helped us enhance the discussion in this manuscript section. The sentence on line 328 was modified according to the comment.
Line 370: The expression of genes in the thyroid, including Slc5a5, Tpo, and Dio1 , was observed to be lowest in the BO and BR groups. Statistical analysis confirmed that these differences were significant (p < 0.05), suggesting a clear impact of excess dietary iodine on the activity of these genes.
Comment 5:
Line 378 – how long were the rats fasted? Response:
Response:
Rats were fasted for 12 hours. This information was added to the manuscript -line 429
Comment 6:
Line 379 – I assume “plain” means tubes without anticoagulant?
Response:
The sentence was changes into “.....test tubes without anticoagulant"
Comment 7:
Line 380 – additional information regarding how the blood was handled should be added, for example, how long was the sample allowed to clot? Was it left at room temperature to clot or on ice? Was it centrifuged at room temperature?
Response:
Blood samples were left at room temperature for about 2 hours to clot. This information was added to the text - line 431.

Round 2
Reviewer 1 Report
Comments and Suggestions for Authors
All suggestions and comments were attended.

Reviewer 3 Report
Comments and Suggestions for Authors
Thank you for the revisions to the manuscript.